# How Do We Define and Measure Optimal Care for Cancer Survivors? An Online Modified Reactive Delphi Study

**DOI:** 10.3390/cancers13102299

**Published:** 2021-05-11

**Authors:** Karolina Lisy, Lena Ly, Helana Kelly, Melanie Clode, Michael Jefford

**Affiliations:** 1Department of Health Services Research, Peter MacCallum Cancer Centre, Melbourne, VIC 3000, Australia; Karolina.lisy@petermac.org; 2Australian Cancer Survivorship Centre, Peter MacCallum Cancer Centre, Melbourne, VIC 3000, Australia; helana.kelly@petermac.org (H.K.); melanie.clode@mcri.edu.au (M.C.); 3Sir Peter MacCallum Department of Oncology, Faculty of Medicine, Dentistry, and Health Sciences, University of Melbourne, Melbourne, VIC 3010, Australia; 4Faculty of Medicine, Melbourne School of Population and Global Health, Dentistry, and Health Sciences, University of Melbourne, Melbourne, VIC 3010, Australia; lyl2@student.unimelb.edu.au

**Keywords:** cancer survivor, survivorship, quality of healthcare, quality of life, long-term treatment effects, models of care, Delphi study, quality criteria

## Abstract

**Simple Summary:**

We aimed to determine ‘what quality criteria do survivorship experts consider to be important in achieving optimal cancer survivorship care?’. An online modified reactive Delphi study was conducted over two rounds. Experts included consumers, clinicians, researchers, policymakers and quality and accreditation groups. In round 1, experts ranked the importance of 68 criteria, derived from the international literature across three domains: Policy, process and outcome. In round 2, experts ranked their top 10 criteria per domain. These 30 items formed the Quality Framework. A consensus meeting considered priority items from the Quality Framework and the feasibility of data collection. Prioritized items focused on having a policy supporting provision of survivorship care; the existence of a survivorship program (policy); appropriate processes to assess survivors’ needs following treatment and to stratify to appropriate models of care (process); and reporting survivors’ patient-reported outcomes, quality of life and survival rates (outcome).

**Abstract:**

This research sought to answer the question ‘what quality criteria do survivorship experts consider to be important in achieving optimal cancer survivorship care?’. An online modified reactive Delphi survey consisting of two rounds was completed with experts including consumers, clinicians, researchers, policymakers and quality and accreditation professionals. Survey items were based on international literature and considered three domains: Policy, process and outcome. In round 1 (R1), experts ranked the importance of 68 criteria on a five-point Likert scale. Criteria were retained if scored 4 (important) or 5 (very important) by >75% participants. In round 2 (R2), experts ranked top 10 criteria per domain. Response rates were 79% (70/89) and 84% (76/91), respectively. After R1, six criteria were removed and six were added. From R2, ten items for each domain were retained. These 30 items formed the Quality Framework. A consensus meeting considered priority items from the Quality Framework and feasibility of data collection. Prioritized items included having a policy on survivorship care; the existence of a multidisciplinary survivorship program (policy); appropriate processes to assess survivors’ emotional, psychological and physical needs following treatment and stratification to appropriate models of care (process); and collecting patient-reported outcomes, quality of life and survival rates (outcome).

## 1. Introduction

With improvements in the detection and treatment of cancer, the number of people living beyond cancer continues to grow worldwide [1]. Living beyond cancer remains a challenge for many survivors who may experience persisting physical and psychosocial issues, as well as late effects, including the risk of developing a new cancer and risk of premature death [2,3,4,5]. Current cancer survivorship care is suboptimal, leaving substantial proportions of survivors with persistent symptoms and unmet needs, and insufficient support to return to optimal functioning [2,3,4,5]. Provision of care in Australia is not adequately tailored to individual circumstances and needs, nor is there adequate support for survivors to improve their wellbeing [3]. Care is frequently poorly coordinated and there is an absence of clear responsibility for delivering follow-up care between oncology and primary care providers [4,5]. There is widespread recognition that new models of care are needed to improve outcomes for survivors [1,5,6,7]. Progress in the development of these models may be hindered by the limited availability of measures to assess the quality of care delivery and to assess survivorship outcomes [6,8].

The US Institute of Medicine (IOM, now the National Academy of Medicine) 2001 publication ‘Crossing the Quality Chasm’ proposed that quality health care should be safe, effective, patient-centered, timely, efficient and equitable [9]. Building on these principles, the landmark 2006 report, ‘From Cancer Patient to Cancer Survivor: Lost in Transition’ suggested a shift in the focus of existing survivorship care models from a dominant emphasis on surveillance for disease recurrence to a broader focus, adding the prevention of recurrent and new cancers, as well as late effects; interventions to deal with the consequences of cancer and its treatment, and effective coordination between specialists and primary care providers [1,6]. The IOM report also recommended development of public/private partnerships and quality assurance programs to monitor and improve the care that all survivors receive [1]. Concerning a focus on the quality of survivorship care, the American Society of Clinical Oncology, in its 2013 statement, recommended standardized ways to monitor and improve quality care, encouraged the implementation of quality improvement programs and advised the development of quality measures, to monitor and improve the care that all survivors receive [10]. 

Nekhlyudov et al. (2019) developed a quality survivorship framework aligning with the IOM components, more explicitly calling out a focus on health promotion and disease prevention [8]. The report also suggested the development of recommendations and assessments to improve the quality of survivorship care [8]. Previously, Malin, Sayers and Jefford (2011) proposed that better outcomes for survivors might be achieved by improving the quality of policies and processes of survivorship care [6].

There is a growing focus on improved models of care for cancer survivors in the Australian state of Victoria, as well as a recognition that to improve care, it is imperative to understand what constitutes optimal survivorship care and to develop measures to assess this [11,12,13]. There remains, however, a lack of consensus on metrics to assess the quality of survivorship care or indeed agreed processes and process structures to facilitate optimal care. This work aimed to develop consensus-based quality criteria for optimal survivorship care in Victoria, Australia. The principal research question was ‘what quality criteria do survivorship experts consider to be important in achieving optimal cancer survivorship care?’

## 2. Materials and Methods

The study employed an online modified reactive Delphi technique, a structured consultation process that obtains expert opinions to establish consensus over a series of rounds [14,15]. Although there are no universally accepted requirements or recommendations for conducting Delphi studies [14], the technique has been previously used to reach consensus on the development of quality indicators [14,16,17]. This was a modified Delphi study as we decided a priori to limit data collection to two survey rounds, and to then schedule a consensus meeting to discuss next steps, based on feasibility and relative priorities of identified criteria. The study was reactive as it required experts to rate the importance of a pre-developed list of items (rather than require participants to develop items), and also to suggest further items and/or modify existing items [14].

### 2.1. Initial Survey Development 

The development of the round 1 (R1) Delphi survey was informed by results of an environmental scan of international published and unpublished literature (search current to March 2020) that sought to understand: (1) What are the key components that contribute to optimal survivorship care; and (2) what measures are available to monitor or assess survivorship care. Full methods for the scan and scan results are reported elsewhere [18]. Briefly, the scan employed a systematic search of PubMed for published literature, as well as the review of 42 Australian and international organizational and governmental websites, and consultation with international survivorship experts. Overall, 40 documents were included in the scan. Data regarding components of survivorship care and measures used to assess care were extracted verbatim. We were also guided by items proposed by the survivorship quality framework developed by Nekhlyudov and colleagues [8]. Data were organized by the research team into three categories based on concepts that were being described and/or measured: Policy (principles and procedures guiding an organization’s capacity and structure to provide survivorship care [2,19]), process (an organization’s capacity to deliver survivorship care through its embedded processes [6]) and outcome (how to measure the impacts or effects of survivorship care [6]). Design of the R1 survey retained these three categories, with putative quality indicators developed from the measures and components identified in the scan. Items (quality indicators) within the R1 survey underwent multiple rounds of review and revision by the research team to ensure clarity of each item and ensure all items were captured without duplication. Similarly, the design of the survey, including participant instructions, response scales and free text boxes, underwent multiple rounds of review, revision and pilot testing by the research team.

### 2.2. Participants

A purposive sampling approach was adopted to recruit Australian and international participants between May and September 2020. Given that the ultimate intention of this work is implementation in Victorian health care services, the study deliberately oversampled experts from Victoria. Participants were experts in cancer survivorship, where an expert was defined as someone who: (a) Had received or experienced cancer survivorship care; (b) delivers cancer survivorship care as a component of their work; (c) has been or is currently involved in survivorship care research; (d) is involved in policy, administration or management of any aspect regarding cancer survivorship; and/or (e) is involved in quality or accreditation activities in cancer care. Experts were identified from the following stakeholder groups: Adult cancer survivors; multidisciplinary health care professionals (including oncologists, general practitioners, allied health and specialist and primary care nurses); survivorship/health service researchers; policymakers; administrative and managerial staff, and quality and accreditation staff. Experts were sourced from professional networks of the research team, the study’s steering committee and consumer organizations. Professional participants were invited to opt-in to the study by email. Consumer representatives with prior survey and/or research experience, thereby their survivorship experience would not be limited to their own personal experience were recruited via their affiliation with cancer organizations (Peter MacCallum Cancer Centre, Bowel Cancer Australia, Breast Cancer Network Australia, Prostate Cancer Foundation Australia, Victorian Comprehensive Cancer Centre, Cancer Council Victoria), with a member of the research team contacting each organization to request consumer participation in the study and following the required process for that organization. 

### 2.3. Delphi Study

Surveys were administered using Research Electronic Data Capture (REDCap), a secure web-based survey and database development and management platform. Participants were sent an email invitation containing a link to each survey; surveys were open for two weeks, and reminder emails were sent to non-responders at one week. The landing pages for both R1 and round 2 (R2) surveys contained information including the purpose of the study, requirements for participation and ethical considerations. Participants were required to tick a box indicating that they understood the purpose of the study and what participation involved and provided their consent to participate. The R2 survey was drafted in the week following completion of R1 and analysis of R1 survey results, with email invitations for R2 circulated one week following R1 completion. Experts were not required to have completed R1 to be invited to R2 of the Delphi study. 

Basic demographic data were collected at both R1 and R2 regarding each participant’s role, age, gender, location of work (or residence for survivors) and years of experience in cancer survivorship. 

#### 2.3.1. Round 1 Survey

The R1 survey asked experts to consider quality criteria for survivorship care that were organized into three domains of policy, process and outcome, and to rate the importance of each criterion using a 5-point Likert scale (1-not at all important, 2-not important, 3-somewhat important, 4-important, and 5-very important). Important was defined as being “a core component in achieving quality survivorship care and can be used to measure the quality of cancer survivorship care”. An optional text box was added after each quality criterion to allow participants to provide comments on the items. A text box was also added at the end of each domain to allow participants to list any additional quality criteria that they felt were missing from the list.

#### 2.3.2. Round 2 Survey

In R2, experts were asked to evaluate the revised list of quality criteria developed from R1 and rank what they believed to be the 10 most important criteria within each domain (policy, process and outcome) in order of importance from 1 to 10. Participants were asked to mark the most important criterion as ‘1′, the second most important with ‘2′, and so on. When ranking the criteria, participants were asked to consider the importance of each criterion in optimal survivorship care rather than considering the practical aspects or feasibility of collecting the data.

### 2.4. Data Analysis

#### 2.4.1. Round 1 Survey

R1 results were analyzed using descriptive statistics, including the mean rating and standard deviation for each item, as well as total number and percentage of respondents who rated a criterion as important (score 4-5) on the Likert scale. The survey applied the following consensus thresholds for R1 to determine whether items would proceed to R2: The item was retained if >75% experts considered the item to be important (score 4-5); the item was removed if <50% experts considered the item important. Per the measurement standards applied by Reeve et al. [20], retention of criteria that were considered important by ≥50% but ≤75% of experts was discussed by the research team. The research team considered the strength of evidence supporting the proposed item before deciding whether to retain or remove items.

#### 2.4.2. Round 2 Survey

Data were analyzed for each domain separately. Survey responses were included in the analysis for R2 if participants provided at least 7 out of 10 rankings in at least one domain. Data were analyzed using descriptive statistics including average score, standard deviation, as well as the total number of experts who ranked each criterion in their top 10, and the number of times each item was ranked 1, 2 or 3. Average scores were calculated for each item by reverse-scoring rankings such that an item ranked 1 received 10 points, an item ranked 2 received 9 points, and so on; unranked items were assigned a score of 0. Criteria were then placed in order of importance based on the average scores, and a putative Victorian Quality Cancer Survivorship Care Framework (Quality Framework) was developed by listing the top 10 criteria within each domain.

#### 2.4.3. Consensus Meeting

A consensus meeting was convened by the research team in September 2020 via Zoom video-conferencing to present and receive feedback on the Quality Framework and to discuss its operationalization within Victoria. Attendees for the consensus meeting were selected from amongst the Delphi participants based on their knowledge of factors such as (i) the practical aspects of survivorship care delivery, (ii) the feasibility of data collection in the Victorian setting, (iii) organizational factors within the Victorian healthcare system, (iv) likely funding and staff allocation to progress work relating to the Quality Framework in the following 12–24 months and (v) alignment with government priorities. During the meeting, experts were asked to consider and discuss which quality criteria are a priority, are feasible and achievable to progress in Victoria in the next 12–24 months given a realistic budget. Two online polls were conducted using Zoom. Firstly, participants were asked to assess whether the Quality Framework captured key elements of quality cancer survivorship care in Victoria. Secondly, participants were presented with a summary of the meeting outcomes and asked whether they agreed that these represented an appropriate way to progress the quality of survivorship care in Victoria over the next 12–24 months. Results from the meeting were summarized in a report and circulated to participants.

## 3. Results

### 3.1. Participants

For R1, a total of 89 experts opted-in and were sent a link to the R1 survey, and 70 experts (79%) completed the survey. For R2, a total of 91 experts opted-in, and 76 experts (84%) completed the survey, with two participants’ responses excluded due to missing data for a total of *n* = 74. Participants were mainly in clinical roles or were survivors or researchers. On average, experts reported 14–15 years’ of survivorship experience. See Table 1.

### 3.2. Round 1

The original list of 68 quality criteria including 17 policy, 30 process and 21 outcome items were ranked in order of importance based on the total number and percentage of experts who rated the items four or five on the Likert scale. Overall, 6 criteria were added (3 in policy, 3 in outcome), 6 were removed (3 from policy, 1 from process and 2 from outcome), and 18 were modified (4 from policy, 11 from process, 3 from outcome). Details are included in Appendix A. Rankings from the R1 survey are shown in Appendix A.

Additional criteria included having a policy that considers transitions in survivorship care; outlines the role of consumers in survivorship programs and considers public reporting and dissemination of survivorship outcomes (policy); the collection of data on health professionals’ views of survivorship care and their own wellbeing, and the number and characteristics of survivors lost to follow-up, as well as the number of referrals made for survivors (outcome).

Items that were removed included: ‘policy that describes how survivorship care data is collected and stored’ (70% rated 4 or 5); ‘policy that requires and records relevant staff training in survivorship care’ (66% rated 4 or 5) ‘policy that describes how survivorship care data is used for policy development and practice improvement’ (64% rated 4 or 5) (policy); ‘cancer survivors are provided with access to screening services for exposure to infectious disease or conditions (e.g., Hepatitis B)’(56% rated 4 or 5 (process); the organization has a process to ‘collect data on the average waiting time for follow-up services’ (71% rated 4 or 5) and the organization has a process to ‘collect data on survivors’ vaccination rates’ (46% rated 4 or 5) (outcome). 

Most free-text responses obtained from experts concerned the wording or clarity of select criteria. The wording and language of 18 criteria were revised based on experts’ feedback from the R1 survey (detailed in Appendix A). For example, one common revision of criteria was regarding the addition of ‘as necessary’ to emphasize that assessments, referrals and advice should be provided to cancer survivors only when appropriate. To ensure consistency of wording between criteria in the process domain, several other revisions involved changing the stem to ‘survivors are provided access to…’.

### 3.3. Round 2

Responses were analyzed from a total of 74 participants. Respectively, 99% (*n* = 73/74) of experts had sufficiently ranked their top 10 criteria in the policy domain and 95% (*n* = 70/74) in each of the process and outcome domains. Quality criteria within each domain were then ordered based on average scores (Table 2, Table 3 and Table 4), forming the Quality Cancer Survivorship Care Framework (Table 5).

The highest-ranking quality criterion in the policy domain concerned having a policy to describe a framework for the provision of survivorship care (Table 2). Sixty-two out of seventy-four (83.4%) participants ranked this item in their top 10 and the highest average score of 7.78 out of a maximum score of 10 was obtained. Other highly endorsed items included having policies that require the establishment of a survivorship program onsite or by referral, and that outline the team of multidisciplinary health professionals included in the survivorship program. Prioritized criteria in the process domain included having a process to assess the emotional, psychological, and physical effects of cancer following treatment, as well as ensure that survivors are stratified to appropriate models of care based on factors such as their current needs and predicted risks. Top-ranking criteria in the outcome domain were patient-centric items involving collecting data on survivors’ patient-reported outcomes (PRO), quality of life (QoL) and survival rates.

### 3.4. Consensus Meeting

The final Victorian Quality Cancer Survivorship Care Framework with 30 criteria across three domains was presented to the consensus meeting attendees for discussion regarding priority actions and feasibility of data collection. The meeting was attended by a total of 19 experts (of 20 invited), comprising of Victorian Department of Health staff (*n* = 5), cancer survivors or consumers (*n* = 2), program managers (*n* = 3), clinicians (*n* = 6) and researchers (*n* = 3). Experts were asked to consider the ten items in each domain, and whether any approach could address multiple items. In the policy domain, there was general agreement among experts to progress work on developing a generic survivorship policy, which can be tailored to health organizations and health services, to document the organization’s approach (including a minimal expectation) around survivorship care. The scope of the policy will likely be broad but may include several of the criteria from the policy domain. Experts also agreed on grouping three broad areas in the process domain: (1) Progress guidance and tools around needs assessment, which may consolidate a number of relevant quality criteria and incorporate assessment of physical, psychological/emotional and practical/social issues into the one assessment, (2) consideration of appropriate treatment and referral pathways, in response to identified need and (3) stratification to appropriate models of care, based on needs and predicted risks. The collection of a standardized set of patient-reported outcomes was prioritized in the outcome domain. These may include specific outcomes of quality of life, functional capacity and return to previous functioning (e.g., work or study).

Experts also discussed leveraging existing data collection processes, including the Victorian Healthcare Experience Survey and collection of survival data through the Victorian Cancer Registry. Polling was completed at the end of the session, and 17 of 19 experts remained. The first poll showed that 100% (*n* = 17/17) of the experts were satisfied that the framework determined the key elements of quality survivorship care in Victoria. The final poll also revealed that there was universal agreement (*n* = 17/17) among experts that the summary of the discussion presented at the meeting was an appropriate way forward to progress the quality of survivorship care in Victoria over the next 12–24 months.

## 4. Discussion

Though there are growing calls internationally for improved survivorship care and alternative care models [1,5,6,7], implementation remains challenging, in part due to a lack of clear, measurable indicators that may be used to inform and assess survivorship care delivery and outcomes. There have also been growing recommendations to develop frameworks with embedded measures that consider the quality of survivorship care to guide the redesign of health care organizations [1,6]. This study sought to address this gap through development of a consensus based Victorian Quality Cancer Survivorship Care Framework.

An environmental scan was undertaken to identify components of optimal survivorship care, alongside measures that may be used to assess survivorship care. An online modified reactive Delphi technique was employed to build consensus on criteria to assess the quality of survivorship care in three domains: Policy, process and outcome. The resulting Quality Framework includes 30 criteria to assess quality survivorship care across the three domains, and this study also established consensus on how to operationalize the framework locally in the short term. 

Within the Quality Framework policy domain, having a policy that describes a framework for the provision of survivorship care, requires the establishment of a survivorship program onsite or by referral and outlines the team of multidisciplinary health professionals included in the survivorship program were the most highly endorsed by experts. Importantly, having a survivorship program with defined roles is consistent with the American’ College of Surgeon’s Commission on Cancer (CoC) accreditation standard 4.8 [21]. This new standard requires the development and implementation of a survivorship program designed to improve care for cancer survivors treated with curative intent. The standard requires the program to include the following features, many of which are present in our Victorian Quality Cancer Survivorship Care Framework: A designated survivorship coordinator, who reports to the site’s cancer committee and multidisciplinary team members including physicians, advance practice providers, nurses and allied health professionals; the survivorship program must provide at least three services per year (e.g., provide treatment summaries or survivorship care plans, rehabilitation services, psychological services, referrals to expert cardiology, fertility services, etc.) to cancer survivors; and monitor and evaluate their performance via the cancer committee. This reflects a shift in accreditation for the US, where the accreditation standard was previously dependent on a threshold of survivorship care plans provided to cancer survivors. This continues to be encouraged but is not a requirement of the standard. Reflecting on this change, Blaes et al. note that this change is supported by ASCO, especially the emphasis on care across the survivorship continuum and the focus on the process of delivering care [22]. Strong support for the development of a generic survivorship policy was also prioritized at the consensus meeting as an immediate way to operationalize the Quality Framework. The research team have begun work to progress this for Victorian health care organizations.

A recent publication by Dunn et al. (2020) used a Delphi approach to describe men’s current prostate cancer survivorship experience. Six domains were identified, including care coordination [23]. Their study also suggests that the delivery of survivorship care should involve multidisciplinary health professionals including clinical teams, primary care clinicians, nurses, allied health professionals, community-based health and welfare services. Care coordination and communication of patient information and care plans between these health care providers, as well as the provision of referrals to community support groups for survivors can help to foster patient-centered care [23]. 

Within the Quality Framework process domain, assessment of emotional, psychological and physical effects of cancer were identified as important criteria. Our center has previously published work on needs assessment tools specific to the post-treatment phase [24]. Five needs assessment tools were identified; none covered all areas of unmet needs (physical, emotional, lifestyle/information, family/relationships, sexual and cognition), and in terms of their psychometric properties, none demonstrated evidence of all required criteria of validity and reliability. The lack of quality, comprehensive tools was highlighted and underscored by the lack of guidance on the implementation of needs assessment of cancer survivors. Identification and reassessment of symptoms using history and/or validated instruments is important to determine effective strategies and interventions to help survivors manage late effects of cancer and its treatment [1,8]. Practical guidance and tools continue to be lacking in Australia. 

This study also revealed risk stratification/personalized care as an important element of optimal survivorship care. Risk stratification is a personalized care approach that involves assignment of survivors to specific models of care based on factors including diagnosis and treatments, indicated needs and predicted risks [25]. This was reflected across documents included from Australia, Ireland, the UK and USA in the environmental scan. Indeed, survivors with low risk of late effects may only need post-treatment follow-up care in the primary care setting, whereas survivors with multiple complex needs may require care provided by oncologists and a multidisciplinary team of specialists. Shared care between primary care providers and oncologists may be suitable for survivors with moderate levels of risk [25]. Awareness of the importance of different models of care, based on predicted risks and identified needs (stratified pathways of care), is an important finding from this study, which reflects emerging thinking. We suggest this item may not have been prioritized a decade ago in Victoria, likely reflecting increased awareness of international approaches to survivorship care (particularly promotion of personalized stratified follow up (PSFU) in the UK [26,27]). In the UK, survivors are stratified to either supported self-management with remote monitoring, or scheduled follow up, in person or via telephone. Those who are stratified to self-management are provided with information about signs and symptoms that may indicate recurrence; have rapid re-access to their cancer team if needed; receive regular surveillance scans or tests, as well as personalized care and support to enable self-management. Additionally, survivors receive end of treatment summaries, a primary care cancer review and health and wellbeing information and support. Victoria may look to the UK experience for both guidance on needs assessment tools and the integration of stratified pathways of care within existing cancer systems and services, including the implementation approach of beginning with a limited number of cancer types first, then expanding to others. 

In cancer care, cure and survival rates are important outcomes. Notably, within the outcome domain of the Quality Framework, experts ranked QoL and other PRO data more highly than survival rates (e.g., one and five year survival) as measures of quality survivorship care. This is an important finding, highlighting not only the value of an expert consensus-building process, but also a shift in emphasis towards ensuring people live well after cancer. QoL and PRO measures have been recommended previously [28,29,30]. Part of the difficulty in adopting them has related to a lack of agreement on which measures to use. Ramsey et al. (2020) have contributed important data in this regard [31]. They conducted focus groups with cancer survivors and a review of instruments used to assess survivors’ QoL, alongside a two-round Delphi survey and consensus meeting to define a core set of patient-reported outcomes to be used in population level survivorship research as outcome measures. The core set contained 12 domains, including assessing survivors’ functioning in daily social and occupational activities alongside QoL as a broad measurement of satisfaction, physical, emotional and social wellbeing [31]. 

The UK has begun national collection of PROs. The NHS England commenced the cancer quality of life survey as a pilot in 2017, and more broadly from 2020 [32]. The QoL assessment is generated through two questionnaires, EQ-5D (generic questionnaire rating mobility, self-care, activities, pain, anxiety on a visual analogue scale) and the EORTC QLQ-C30 (cancer-specific questionnaire rating physical, cognitive, emotional, social, finance, symptoms, global health). Cancer survivors receive the cancer quality of life survey as a one-off survey approximately 18-months post cancer diagnosis. Initially only provided to certain cancer groups, by 2022, most survivors will be included in this nationwide survey. Victoria could undoubtedly learn from the UK experience in QOL tools and implementation at scale. 

This Delphi study builds on Nekhlyudov et al.’s work to develop a Quality Framework for cancer survivorship care, using it “as a road map for future efforts aiming to systematically measure and improve cancer survivorship care quality in clinical care, research, and policy” [8] (p 1127) as the authors intended. We have been able to leverage their important work, operationalize it for our context and prioritize ‘what’s next?’ for Victoria as we seek to improve survivorship care. Most notably, this study proposes process and outcomes measures, as well as priority and feasible immediate next steps for Victorian health care services, identified through the consensus meeting. This adds unique translational value to the research; it provides an immediate way forward and supports the application of the Quality Framework into practice. 

The extent to which a framework can be disseminated and adopted depends on the availability of resources across health care, social and community systems that deliver survivorship care. The meeting also addressed comments regarding the practical aspects of collecting the data, which the Delphi study had intentionally avoided. Indeed, experts agreed to progress work on the collection and reporting of survival and recurrence rates, which remains part of the population-based Victorian Cancer Registry’s responsibility, as well as the strongly endorsed items on measuring QoL and PROs. Importantly, the Victorian Cancer Plan 2020–2024 [12] specifically names this work in action area 4.3, committing to developing and implementing a framework that defines and measures quality survivorship care by 2024 [12], reflecting this study’s contribution to cancer policy and future implementation. The Victorian State Government has recognized for some time that metrics are important, but there was a lack of clarity about how to define and measure quality survivorship care, and this study not only informs which quality metrics are important, but provides clarity on which to prioritize, based on expert consensus. Importantly, through the Victorian Cancer Plan [12], there is now a policy imperative to take action. 

### 4.1. Strengths

The inclusion of a broad range of Australian and international participants with considerable expertise in cancer survivorship ensured that diverse views on survivorship care from all relevant stakeholder groups were represented in the results. The study’s purposive oversampling of Victorian participants served to improve local representation within the study sample, increasing the likelihood that study outcomes were responsive to the views of those who will be utilizing the Quality Framework. To further enhance the acceptance of the framework [14], the study’s consensus meeting involved discussion amongst Australian survivors and consumers, and health care professionals, including clinicians, researchers and policymakers. Importantly, attendees included Victorian Department of Health staff whose expertise and influence in cancer policy will be critical to drive support for operationalization of the framework in Victoria.

Quality criteria were initially developed based on a current and comprehensive environmental scan incorporating Australian and international published and unpublished literature. Moreover, the validity of the framework was strengthened by the study’s use of a Delphi technique, where the content was determined by the wide-ranging perspectives of cancer survivorship experts [33]. The consensus-building process also provided experts with an opportunity to rank and prioritize the quality criteria based on the incorporation of other experts’ anonymous feedback and additional suggestions after the first-round survey. Typically, consensus among participants is built iteratively with each additional round in Delphi studies. An advantage of limiting our study to two survey rounds is reducing potential bias due to participant fatigue and attrition that is often associated with increasing Delphi rounds [15,31].

### 4.2. Limitations

The ultimate goal of this work was to develop a Quality Cancer Survivorship Care Framework for implementation in Victoria, Australia; as such, results may not apply to other Australian jurisdictions or international settings with varied delivery and funding of health care systems. We also recognize that the Quality Framework may not represent the private health care sector. In light of these limitations, further work may explore the adaptability of this framework beyond the Victorian public health care system. This study did not define the meaning of individual terms used to describe specific quality criteria, for example ‘survivorship program’, as this was considered outside of the scope of the present work. Therefore, experts may have had different views and interpretations of some of the quality criteria proposed in this study. Future work may include delineating precise meanings of terms based on context and setting of implementation. Given that there is no universally accepted methodology for undertaking the Delphi technique, this study referred to the consensus thresholds and number of survey rounds used in select published Delphi studies (for example [15,20]). Therefore, it is unclear whether the present results would have been affected if different thresholds and methods had been applied [31].

## 5. Conclusions

This study used a consensus-building process to develop a framework to improve the quality of survivorship care in Victoria that will likely be of interest to survivors and consumers, researchers, clinicians and policymakers. Results of this study provide the first steps to define and measure optimal survivorship care across Victorian health care organizations. We have been able to operationalize the Quality Framework, determining the next steps in the Victorian setting. These include supporting Victorian health care services to implement a policy that describes a framework for survivorship care; focusing on screening survivors for symptoms and unmet needs and linking them to appropriate services; and progressing the use of outcome measures, including measures of QoL and other PROs. 

Future work can focus on defining the parameters of each quality criterion in the Quality Framework, alongside monitoring the extent to which implementation of the framework impacts survivors’ health and wellbeing beyond treatment, as well as the health care system. 

## Figures and Tables

**Table 1 cancers-13-02299-t001:** Demographic data from experts in both survey rounds.

	Round 1 (*n* = 70)		Round 2 (*n* = 74)	
DEMOGRAPHICS	*n*	%	*n*	%
Gender				
Female	53	76	57	77
Male	17	24	17	23
Living in Australia				
Victoria	48	69	52	70
New South Wales	5	7	7	9
Queensland	3	4	3	4
South Australia	3	4	3	4
Western Australia	3	4	0	0
Living overseas				
USA	3	4	3	4
Canada	1	1	2	3
Denmark	1	1	1	1
Japan	1	1	1	1
Netherlands	1	1	1	1
UK	1	1	1	1
Expertise				
Survivor, carer, consumer advocate	15	21	15	20
Researcher	14	20	13	18
Allied health professional	9	13	11	15
Medical oncologist, radiation oncologist, or hematologist	8	11	10	14
Administrative, project or managerial staff	8	11	7	9
Policymaker	4	6	6	8
Specialist nurse	4	6	4	5
Other	3	4	3	4
Surgeon	2	3	3	4
General practitioner	3	4	2	3
Completion of round 1 survey				
Yes	N/A	N/A	64	86
No	N/A	N/A	10	14
	*M (SD)*		*M (SD)*	
Age (years)	52.69 (11.05)		51.91 (10.47)	
Years of experience in cancer survivorship	14.92 (8.60)		14.11 (8.65)	

**Table 2 cancers-13-02299-t002:** Scoring and ranking of quality criteria in the policy domain (results from Round 2 survey).

Rank	The Organization Has a…	Average Score	SD	*N* Ranked in Top 10 (%)	*N* Ranked #1 (%)	*N* Ranked #2 (%)	*N* Ranked #3 (%)
1	policy that describes a framework for the provision of survivorship care	7.78	3.58	62 (83.8%)	38 (51.4%)	14 (18.9%)	2 (2.7%)
2	policy that requires the establishment or existence of a survivorship program either on-site or by referral	5.23	3.71	59 (79.7%)	13 (17.6%)	6 (8.1%)	7 (9.5%)
3	policy outlining the team of multidisciplinary health professionals included in the survivorship program	4.16	3.52	51 (68.9%)	3 (4.1%)	7 (9.5%)	9 (12.2%)
4	policy for the collection of data on survivors’ experiences of survivorship care (e.g., satisfaction with care)	4.01	3.45	52 (70.3%)	2 (2.7%)	3 (4.1%)	12 (16.2%)
5	policy that outlines the role of consumers in the design, evaluation and reporting of survivorship programs	3.59	3.27	52 (70.3%)	4 (5.4%)	3 (4.1%)	6 (8.1%)
6	policy for the evaluation of the survivorship program and reporting of progress	3.25	2.92	50 (67.6%)	1 (1.4%)	2 (2.7%)	4 (5.4%)
7	policy on stratifying survivors to appropriate models of care	3.22	3.48	41 (55.4%)	3 (4.1%)	4 (5.4%)	4 (5.4%)
8	policy outlining the provision of needs assessment tools for survivors at certain time points post-treatments	3.19	3.38	44 (59.5%)	2 (2.7%)	4 (5.4%)	5 (6.8%)
9	policy that has a senior (executive) role identified as the organizational survivorship care champion	3.19	3.51	40 (54.1%)	1 (1.4%)	7 (9.5%)	8 (10.8%)
10	policy that describes the process of survivorship care reporting within an organizational reporting framework	2.92	3.78	31 (41.9%)	2 (2.7%)	10 (13.5%)	4 (5.4%)
11	policy for the provision of support services to survivors with special needs and from diverse cultural backgrounds (e.g., interpreters)	2.60	2.43	46 (62.2%)	0 (0%)	0 (0%)	1 (1.4%)
12	policy that considers transitions in survivorship care (e.g., from pediatric to an adult care setting, acute to survivorship care)	2.29	2.59	40 (54.1%)	0 (0%)	0 (0%)	2 (2.7%)
13	policy that documents survivorship care reporting requirements to a relevant organizational executive committee	2.10	3.36	26 (35.1%)	0 (0%)	8 (10.8%)	1 (1.4%)
14	policy that requires survivorship-focused information to be available in other languages or in a different format for low literacy readers	1.96	2.51	36 (48.6%)	0 (0%)	0 (0%)	3 (4.1%)
15	policy for the collection of data on carers’ experiences of survivorship care (e.g., satisfaction with care)	1.85	2.74	31 (41.9%)	1 (1.4%)	0 (0%)	2 (2.7%)
16	policy that documents survivorship care reporting requirements to a government agency	1.81	3.09	26 (35.1%)	2 (2.7%)	4 (5.4%)	3 (4.1%)
17	policy around public reporting and dissemination of survivorship outcomes	1.29	2.32	22 (29.7%)	1 (1.4%)	0 (0%)	0 (0%)

**Table 3 cancers-13-02299-t003:** Scoring and ranking of quality criteria in the process domain (results from Round 2 survey).

Rank	Cancer Survivors Are…	Average Score	SD	*N* Ranked in Top 10 (%)	*N* Ranked #1 (%)	*N* Ranked #2 (%)	*N* Ranked #3 (%)
1	assessed for emotional and psychological effects of cancer and its treatment (e.g., anxiety, depression)	4.09	3.67	42 (56.8%)	2 (2.7%)	7 (9.5%)	8 (10.8%)
2	assessed for physical effects following primary treatment (e.g., pain, fatigue, weight loss or gain)	3.91	4.05	37 (50%)	7 (9.5%)	7 (9.5%)	7 (9.5%)
3	stratified to appropriate models of care based on factors such as current needs and predicted risks	3.89	4.38	35 (47.3%)	13 (17.6%)	6 (8.1%)	6 (8.1%)
4	provided with treatment or referrals to manage physical effects of cancer and its treatment	3.60	3.63	39 (52.7%)	3 (4.1%)	4 (5.4%)	7 (9.5%)
5	assessed for practical and social effects of cancer and its treatment (e.g., relationship difficulties, financial challenges, education and employment/return to work)	3.46	3.15	45 (60.8%)	2 (2.7%)	1 (1.4%)	5 (6.8%)
6	provided access to a survivorship program which addresses the needs of cancer survivors either on-site or by referral	3.09	3.61	37 (50%)	8 (10.8%)	2 (2.7%)	2 (2.7%)
7	assessed for their risk of recurrent or new cancer, including family history (as necessary)	2.97	4.20	27 (36.5%)	11 (14.9%)	4 (5.4%)	3 (4.1%)
8	provided with a survivorship care plan that is shared with their primary care provider and/or other multidisciplinary health professionals involved in their care	2.90	3.83	31 (41.9%)	8 (10.8%)	3 (4.1%)	3 (4.1%)
9	provided with recommendations regarding surveillance for recurrent or new cancers	2.76	3.65	31 (41.9%)	3 (4.1%)	9 (12.2%)	1 (1.4%)
10	provided with treatment or referrals to manage psychosocial effects (e.g., to psychology services)	2.23	2.69	33 (44.6%)	0 (0%)	0 (0%)	3 (4.1%)
11	involved in care planning conversations and provided with a survivorship care plan	2.21	3.56	25 (33.8%)	5 (6.8%)	4 (5.4%)	2 (2.7%)
12	provided with recommendations to reduce the risk of any physical effects (e.g., weight loss, exercise)	1.84	3.05	21 (28.4%)	0 (0%)	3 (4.1%)	3 (4.1%)
13	provided with care which is respectful of and consistent with their goals	1.83	3.33	22 (29.7%)	4 (5.4%)	4 (5.4%)	1 (1.4%)
14	provided access to allied health services (e.g., nutrition, physical therapy, sexual health, rehabilitation, dental and podiatry services)	1.80	3.05	26 (35.1%)	2 (2.7%)	4 (5.4%)	1 (1.4%)
15	provided access to education and resources about the post-treatment phase which meets individuals’ needs, understanding and health literacy	1.64	2.61	29 (39.2%)	0 (0%)	3 (4.1%)	2 (2.7%)
16	provided access to primary care services (e.g., GP visits and testing focused on management of chronic medical conditions, health promotion and disease prevention)	1.43	2.62	22 (29.7%)	0 (0%)	2 (2.7%)	3 (4.1%)
17	assessed for lifestyle behaviors with recommended management, or provided with an appropriate referral (e.g., quit smoking programs)	1.27	2.17	24 (32.4%)	0 (0%)	1 (1.4%)	0 (0%)
18	assessed for their self-management skills and appropriately stratified according to their ability to self-manage with support	1.26	2.47	20 (27%)	0 (0%)	2 (2.7%)	2 (2.7%)
19	provided with advice on medications as appropriate to manage physical, psychosocial effects and/or chronic medical conditions	1.24	2.34	19 (25.7%)	0 (0%)	2 (2.7%)	0 (0%)
20	provided access to remote surveillance programs where appropriate (i.e., having tests undertaken close to home, potentially without the need for clinical visits)	1.09	2.30	16 (21.6%)	0 (0%)	0 (0%)	3 (4.1%)
21	provided the opportunity to participate in research projects including clinical trials	0.86	1.98	16 (21.6%)	0 (0%)	0 (0%)	2 (2.7%)
22	provided access to specialty care services to manage potential late effects, as necessary (e.g., cardiology)	0.84	1.92	15 (20.3%)	0 (0%)	0 (0%)	1 (1.4%)
23	assessed for adherence to recommended strategies to manage consequences of cancer and its treatment	0.83	2.07	14 (18.9%)	1 (1.4%)	1 (1.4%)	0 (0%)
24	provided access to telehealth services (e.g., video-based consultations)	0.79	1.93	13 (17.6%)	0 (0%)	0 (0%)	0 (0%)
25	provided access to age- and gender-appropriate cancer screening or referrals to appropriate cancer screening services (e.g., mammograms)	0.69	1.95	10 (13.5)	0 (0%)	1 (1.4%)	1 (1.4%)
26	provided access to education and resources for their carers, about the post treatment phase which meets their needs, understanding and health literacy	0.66	1.68	13 (17.6%)	0 (0%)	0 (0%)	1 (1.4%)
27	provided support or referral for other medical or chronic conditions which are non-cancer related (e.g., diabetes)	0.64	1.74	11 (14.9%)	0 (0%)	0 (0%)	1 (1.4%)
28	provided with referrals for genetic testing (as necessary) following primary treatment	0.43	1.46	8 (10.8%)	0 (0%)	0 (0%)	1 (1.4%)
29	provided access to advice on vaccinations (e.g., influenza)	0.09	0.72	1 (1.4%)	0 (0%)	0 (0%)	0 (0%)

**Table 4 cancers-13-02299-t004:** Scoring and ranking of quality criteria in the outcome domain (results from Round 2 survey).

Rank	The Organization Has a Process to…	Average Score	SD	*N* Ranked in Top 10 (%)	*N* Ranked #1 (%)	*N* Ranked #2 (%)	*N* Ranked #3 (%)
1	collect data on survivors’ patient-reported outcomes	6.29	3.95	55 (74.3%)	23 (31.1%)	6 (8.1%)	7 (9.5%)
2	collect data on survivors’ quality of life	6.27	3.25	60 (81.1%)	6 (8.1%)	16 (21.6%)	10 (13.5%)
3	collect data on survival rates (e.g., one and five-year survival rates)	4.67	4.55	41 (55.4%)	19 (25.7%)	9 (12.2%)	3 (4.1%)
4	collect data on recurrence rates	3.76	4.08	39 (52.7%)	2 (2.7%)	16 (21.6%)	8 (10.8%)
5	collect data on survivors’ patient-reported experiences of care	3.31	2.85	47 (63.5%)	2 (2.7%)	0 (0%)	3 (4.1%)
6	collect data on survivors’ return to previous functioning (e.g., work, study)	2.99	3.12	39 (52.7%)	1 (1.4%)	4 (5.4%)	1 (1.4%)
7	collect data on survivors’ functional capacity	2.96	3.41	35 (47.3%)	2 (2.7%)	0 (0%)	8 (10.8%)
8	collect data on the diagnosis of new cancers (for survivors)	2.67	3.63	30 (40.5%)	1 (1.4%)	3 (4.1%)	12 (16.2%)
9	collect data on carers’ quality of life	2.49	3.31	31 (41.9%)	0 (0%)	2 (2.7%)	8 (10.8%)
10	collect data on the number of survivors provided with a survivorship care plan	2.39	3.07	36 (48.6%)	3 (4.1%)	3 (4.1%)	1 (1.4%)
11	collect data on the overall cost of care to survivors	2.21	2.64	36 (48.6%)	1 (1.4%)	0 (0%)	1 (1.4%)
12	collect data on the overall cost of survivorship care to the health service	1.84	2.52	30 (40.5%)	0 (0%)	0 (0%)	0 (0%)
13	collect data on the number of survivors who have their needs assessed at certain time points post-treatment	1.84	2.97	29 (39.2%)	3 (4.1%)	2 (2.7%)	2 (2.7%)
14	collect data on survivors’ satisfaction with care	1.77	2.89	24 (32.4%)	2 (2.7%)	1 (1.4%)	1 (1.4%)
15	collect data on the number of survivors receiving guideline-compliant surveillance testing	1.66	2.63	24 (32.4%)	1 (1.4%)	1 (1.4%)	1 (1.4%)
16	collect data on the number of primary care providers provided with a survivorship care plan	1.61	2.60	27 (36.5%)	1 (1.4%)	2 (2.7%)	1 (1.4%)
17	collect data on the number of survivors stratified to different models of care (e.g., survivors self-managing their conditions)	1.54	2.79	26 (35.1%)	2 (2.7%)	3 (4.1%)	1 (1.4%)
18	collect data on the number of health professionals trained to provide survivorship care	1.10	2.18	20 (27%)	1 (1.4%)	0 (0%)	1 (1.4%)
19	collect data on the number and characteristics of survivors lost to follow-up	0.99	1.91	21 (28.4%)	0 (0%)	1 (1.4%)	0 (0%)
20	collect data on health professionals’ views of survivorship care and their own wellbeing	0.70	1.73	11 (14.9%)	0 (0%)	0 (0%)	0 (0%)
21	collect data on survivors’ hospital admissions	0.64	1.82	9 (12.2%)	0 (0%)	1 (1.4%)	0 (0%)
22	collect data on the number of referrals made for survivors	0.56	1.31	15 (20.3%)	0 (0%)	0 (0%)	0 (0%)

**Table 5 cancers-13-02299-t005:** Victorian Quality Cancer Survivorship Care Framework.

POLICY DOMAIN The Organization Has a…	PROCESS DOMAIN Cancer Survivors Are…	OUTCOME DOMAIN The Organization Has a Process to…
1. policy that describes a framework for the provision of survivorship care	1. assessed for emotional and psychological effects of cancer and its treatment (e.g., anxiety, depression)	1. collect data on survivors’ patient-reported outcomes
2. policy that requires the establishment or existence of a survivorship program either on-site or by referral	2. assessed for physical effects following primary treatment (e.g., pain, fatigue, weight loss or gain)	2. collect data on survivors’ quality of life
3. policy outlining the team of multidisciplinary health professionals included in the survivorship program	3. stratified to appropriate models of care based on factors such as current needs and predicted risks	3. collect data on survival rates (e.g., one and five-year survival rates)
4. policy for the collection of data on survivors’ experiences of survivorship care (e.g., satisfaction with care)	4. provided with treatment or referrals to manage physical effects of cancer and its treatment	4. collect data on recurrence rates
5. policy that outlines the role of consumers in the design, evaluation and reporting of survivorship programs	5. assessed for practical and social effects of cancer and its treatment (e.g., relationship difficulties, financial challenges, education and employment/return to work)	5. collect data on survivors’ patient-reported experiences of care
6. policy for the evaluation of the survivorship program and reporting of progress	6. provided access to a survivorship program which addresses the needs of cancer survivors either on-site or by referral	6. collect data on survivors’ return to previous functioning (e.g., work, study)
7. policy on stratifying survivors to appropriate models of care	7. assessed for their risk of recurrent or new cancer, including family history (as necessary)	7. collect data on survivors’ functional capacity
8. policy outlining the provision of needs assessment tools for survivors at certain time points post-treatments	8. provided with a survivorship care plan that is shared with their primary care provider and/or other multidisciplinary health professionals involved in their care	8. collect data on the diagnosis of new cancers (for survivors)
9. policy that has a senior (executive) role identified as the organizational survivorship care champion	9. provided with recommendations regarding surveillance for recurrent or new cancers	9. collect data on carers’ quality of life
10. policy that describes the process of survivorship care reporting within an organizational reporting framework	10. provided with treatment or referrals to manage psychosocial effects (e.g., to psychology services)	10. collect data on the number of survivors provided with a survivorship care plan

## Data Availability

Complete data from this study will be made available from authors on request.

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
