# Peer review of "How Do We Define and Measure Optimal Care for Cancer Survivors? An Online Modified Reactive Delphi Study"

_cancers, 2021, doi:10.3390/cancers13102299_

Round 1
Reviewer 1 Report
How do we define and measure optimal care for cancer survivors? An online modified reactive Delphi study
Manuscript ID: cancers-1188003
Thank you for giving me the opportunity to review this interesting manuscript. It contains a description of the development of the Victorian Quality Cancer Survivorship Framework. The Delphi technique has been used, including two survey rounds and a consensus meeting. It is an interesting study, since it highlights the topics that should be included in cancer survivorship care, according to a broad range of experts. The manuscript is also very well written. However, I do have some points that need some more clarification, mainly with regard to the methods used and the discussion section.
The initial survey development: this is only very briefly explained in the manuscript. Can you elaborate a bit more here? Was the search systematic? How many studies were included? How did you select the three categories (a priori of content bases)?
The survey rounds: Page 4, line 180-183: this sounds a bit strange. Why did you choose this method? And how many factors did you decide on? ‘The strength and extent of supporting evidence considered’ is completely unclear. Please elaborate this step in detail. And in line with this: it is unclear what the results of round 1 were. Please put all results of this survey round in an additional file. Also make sure that you show in this additional file what criteria you as a research team have decided on and what criteria were modified (page 6, line 222).
The consensus meeting: The (development of the) framework is very interesting for both research and practice, but it is unclear what the added value of the consensus meeting is for the manuscript. The link with the framework is also not entirely clear. It seems that the results on page 12 are not in line with the framework (at least not the same wording is used). Either delete the consensus meeting from the manuscript or substantiate why the is of interest for a broader group of researchers and practitioners. I also have some methodological concerns that are related to the consensus meeting:
- How many experts were invited for the meeting?
- On what basis were these experts selected? Now the definitive results are based on experts opinions who were selected by the research team after they have filled out survey round 1 and 2. Were the answers in round 1 and 2 taken into account when inviting the expert for the consensus meeting?
The discussion and conclusion: These can be improved by reflecting on your specific results more comprehensively. For the discussion: what is new? Wat adds this study to research and practice? The discussion is now more a description of the current state of research and practice. For the conclusion: you might have written this conclusion before you actually had the results of your delphi study. Please add what you have found by answering your research question.
In addition, some minor things:
- Please make sure that the tables are correctly aligned (for example table 3, line 16 and 17 / and all numbers high aligned)
- Please add percentages to tables 2, 3 and 4. Or put the total N in these tables.
- Table 4: please mention that the scores of different domains cannot be compared due to a different number of factors per domain.
- Please add percentages were needed in the text, for example: page 12, line 256.
- The abbreviation ‘PRO’ (page 14, line 374) has not been introduced.
- 4.5. might be 2.5.
- How do you define ‘survivorship experience’ for cancer survivors? More experience = longer recall period, so less reliable results, right?
- Please sort table 1 ‘expertise’ correctly (next page is not included now)
- Please elaborate why you think that the experts from other areas of Australia and even from overseas are experts for the Victorian situation.
- Page 2, line 47: please add fatigue.
- Page 2, line 45: reference is missing.
- Page 2, second paragraph: please shorten.
- Please explain ‘modified reactive’ somewhere in the manuscript.
- Page 3, line 116: were the surveys also pilot tested by the target population?
- Page 4, line 150: why not? Was this defined a priori of on the basis of non-response of R1 participants?
- Page 4, line 161: please add the possibility to modify the items. Right?
- Page 4, line 179-180: were these criteria defined a priori?
- Was feedback of the results of round 1 provided in the survey of round 2? This is an important aspect of Delphi studies. If not, please elaborate why not.
- Please elaborate why you choose that all experts were involved in all categories. For example, why are cancer survivors involved in policy components?
- 2 Round 1: please use quotation marks in every paragraph.
- Page 6, line 234: a parenthesis is missing.
I hope these suggestions will help you to further improve this interesting manuscripts. Good luck!
Author Response
Please see the attachment,

Reviewer 2 Report
The authors should be commended on their work in this study. The research design was appropriate for the questions asked and importantly included key stakeholders. The study is well written, insightful and contributes to the literature meaningfully. Cancer survivorship is something we all agree is important but may not agree on what it means. Thus, this work does an excellent job at identifying important elements in achieving optimal cancer survivorship care.
Given the diverse panel of cancer experts included, did the authors look at whether different experts (eg. clinicians vs policy-makers) valued certain criteria more than others?
Reviewer 3 Report
I appreciate the overall quality of the paper and the fact that it contains all important elements including for example precisely defined strenghts and limitations. I don't have any major comments. There are only two things which the authors might find useful to address:
- I think there are too many direct quotations which are not necessary at every occasion.
- What are the direct implications for clinical practice? Based on the results of the study, what are the first things that could/should change?
Round 2
Reviewer 1 Report
Dear authors,
Thanks you for revising the manuscript and for taking my suggestions seriously. I think you did a great job with the revisions. The method and discussion section have significantly improved; the contribution to research and practice (mainly the Victorian practice) is better substantiated and the methods are better described.
I still think that the contribution to international practice and research could be further improved in the discussion section by reflecting more deeply on the international literature. But as said, I'm happy with the revisions and I will advise to approve this revised manuscript.
Good luck with a the next steps in Victoria.
Kind regards,
Michiel Greidanus